# Influences of Intense Physical Effort on the Activity of the Autonomous Nervous System and Stress, as Measured with Photoplethysmography

**DOI:** 10.3390/ijerph192316066

**Published:** 2022-11-30

**Authors:** Ignacio Martínez-González-Moro, Inés Albertus Cámara, María-José Paredes Ruiz

**Affiliations:** Group Research Physical Exercise and Human Performance, University Institute of Aging Research, Mare Nostrum Campus, University of Murcia, 30100 Murcia, Spain

**Keywords:** autonomic nervous system, photoplethysmography, heart rate variability, stress

## Abstract

Background: The autonomic nervous system, which is composed of the sympathetic and parasympathetic nervous system, is closely related to the cardiovascular system. The temporal variation between each of the intervals between the consecutive “R” waves of an electrocardiogram is known as heart rate variability. Depending on the type of activity, both systems can be activated, and also influence the interval between “R” waves. Currently, with advancements in technology and electronic devices, photoplethysmography is used. Photoplethysmography detects changes in the intensity of reflected light that allow differentiation between systole and diastole and, therefore, determines the heart rate, its frequency and its variations. In this way, changes in the autonomic nervous system can be detected by devices such as the Max Pulse^®^. Objective: To determine whether the information provided by Max Pulse^®^ on autonomic balance and stress is modified after intense physical exercise, thereby determining whether there is a relationship with body composition, and also whether there are differences with respect to gender. Materials and Methods: Fifty-three runners (38.9% female) with a mean age of 31.3 ± 8.1 years participated in the study. Two measurements (before and after intense physical effort) were performed with the Max Pulse^®^ device. The flotoplethysmography measurement lasted 3 min, and was performed in the supine position. The exercise test was performed on a treadmill. It was initiated at a speed of 6 and 7 km/h for women and men, respectively. Subjects indicated the end of the test by making a hand gesture when unable to continue the test. Results: Autonomic nervous system activity and mental stress values decreased significantly (*p* < 0.05) in men and women, while autonomic nervous system balance decreased only in women. Physical stress increased (*p* < 0.05) in both sexes. Conclusions: Intense exercise causes changes in variables that assess autonomic nervous system balance and stress, as measured by a device based on photoplethysmography. The changes are evident in both sexes, and are not related to body composition.

## 1. Introduction

Autonomic nervous system (ANS) activity and its actions, especially those related to stress, have been linked to heart rate variability (HRV) [1]. HRV is defined as variations in the time between RR intervals of the electrocardiogram, and this reflects the activity of the autonomic nervous system on cardiac function being a possible indicator for the assessment of stress [2]. The heart has the ability to change the RR interval in different situations. This is due to the close relationship of the ANS and its two components, the sympathetic (SNS) and parasympathetic (PNS) nervous systems with the cardiovascular system [3]. Faced with a state of stress caused by physical exercise, stimulation of the SNS predominates increases in the heart rate (HR), thanks to the release of adrenaline and noradrenaline [4]. Likewise, in a resting situation, the PNS predominates, whose neurotransmitter is acetylcholine [5]. If the ANS is balanced, the PNS tries to decrease the HR, while the SNS does the opposite. This causes fluctuations in the heart rate, which is known as heart rate variability (HRV).

This information has been associated with health; it is considered that a higher HRV indicates a healthier situation, and that the predominance of the SNS is a cause of decreased HRV and lower cardiovascular health [6]. It has also been used as a descriptive factor that relates to different aspects of health and sport [7]. It has been used as a detector of overtraining and fatigue [8], and as an aid to training planning and performance improvement, both in athletes [9] and in cardiac rehabilitation [10]. In addition, other authors have observed that HRV is related to gender [11] and body composition [12].

HRV has classically been assessed from the analysis of the electrocardiogram; then, heart rate monitors began to be used, which, with the help of specific software, different parameters of HRV are calculated [13]. Currently, with the use of photoplethysmography (PPG) and with APPs installed on digital devices (for example: smartphones, watches, activity bracelets and tablets), the use of heart rate monitors is spreading [14,15].

Photoplethysmography is a non-invasive technique that detects changes in pulsatile blood flow between the systole and diastole of the heart. It consists of quantifying the intensity of light reflected by the skin after a light is emitted by an LED spotlight on the wrist, fingertips or in the earlobe [16]. It seems that sensors placed on the finger result in clearer signals than those placed on the wrist [17]. The photodetector analyzes changes in the intensity of reflected light, allowing for differentiation between the systole and diastole and, therefore, determination of the heart rate, its frequency and its variations [18].

There are numerous reviews and studies that have compared modern PPG-based systems with ECGs and heart rate monitors, which generally indicate that they are superimposable, both in children [19] and in older people [20]. The quantification of the number of pulse waves in one minute constitutes the pulse rate (PH), which is equivalent to the heart rate [21]. Thus, the determination of pulse rate variability (PRV) can be considered equivalent to HRV, although they are not necessarily equal [22].

In addition, PRV assessment using PPG can be integrated into other devices for clinical use, in order to increase the quantity and quality of information obtained and provided to patients [23]. One of these devices is “Max Pulse^®^” that, in addition to providing data regarding autonomic balance and stress, also provides data on vascular health by analyzing the morphology of the pulse wave [24]. The software of this equipment processes the information obtained by calculating the HRV parameters, providing information in the form of a punctuation, with the intention of facilitating interpretation [25].

Our aim was to determine if the information provided by Max Pulse^®^ on autonomic balance and stress is modified after intense physical exercise, thereby determining if there is a relationship with body composition, and also if there are differences with respect to gender. Our hypothesis was that the physical exercise performed during a maximum stress test (ST) involves changes in the autonomic nervous system that modify the PRV, and that influences the stress score provided by this device.

## 2. Materials and Methods

### 2.1. Study Design

This was a prospective study that compared the effects of a maximum stress test on the assessment of the ANS by PPG, using a point evaluation system. To do this, we performed two PPG tests, one prior to performing a maximum stress test on a treadmill, and another five minutes after the end of it (effect control). With this, we obtained information about the scores prior to exercise, and after it. All of this is related to the responsiveness of the device. All tests were carried out by the same researcher, between 9.00 a.m. and 12.00 p.m.

### 2.2. Participants

Fifty-three runners, recreational athletes, 38.9% female, participated in this study. Their data included the following: age, 31.3 ± 8.1 years (18–49 years); years of training, 3.9 ± 4.31 (0.5–15), and hours/week, 5.3 ± 2.9 (2–15). Participants practiced running at least two hours a week, and for at least six months prior to the study. They came to our lab in the first half of 2022.

This project was approved by the university’s Research Ethics Committee, and conformed to the Declaration of Helsinki. After being informed of the study design and potential risks, all of the participants signed a written informed consent document, and completed a questionnaire on their medical and sports history.

The inclusion criteria for participating in the program were as follows: being over 18 years old and under 50, and training for running for at least two hours a week, for at least the previous six months. Exclusion criteria included the following: (a) subjects who had known cardiovascular disease or musculoskeletal injuries that could hinder them from performing ST; (b) subjects who were taking growth hormone, testosterone, or other similar anabolic hormonal pharmaceuticals. Moreover, those who regularly took medications that have a direct or indirect effect on the nervous system (e.g., beta-blockers, anxiolytics, antidepressants or neuroleptics); and (c) subjects who did not finish the stress test. Subjects were asked not to compete, train hard, and not to eat a copious meal in the 24 h leading up to the test.

### 2.3. Measurement

Information on the status of the ANS were obtained using a Max Pulse^®^ (Medicore Co., Ltd., Hanam, Republic of Korea); this is a simple, easy-to-use, non-invasive Class II FDA 501k medical device [24]. The device makes a report on the sympathetic/parasympathetic relationship, and physical and mental stress. This information is expressed quantitatively, with its own scoring system obtained by the device’s software. From the analysis of the specific HRV parameters, and by applying an algorithm, Max Pulse^®^ provides a numerical value for each variable. Subsequently, this score is evaluated qualitatively, thereby obtaining information that is easy to interpret for the user (professional, patient or athlete). It is not necessary to have specific training in HRV or physiology to interpret the results (Table 1).

During the measurements prior to the stress test, the subjects were placed on a stretcher in a supine position. Cardiopulmonary auscultation echocardioscopy, resting EKG, blood pressure and basal heart rate were taken. After that, the first PPG was performed, measuring PRV for three minutes while the subjects remained lying down without speaking or moving.

We placed a digital sensor on the index finger of the athlete’s left hand that was connected to Max Pulse^®^ notebook through USB connections (Figure 1). Similarly, PRV was analyzed after the ST.

The ST was performed on a treadmill (model run7411) with a continuous and progressive ramp protocol [26]. Testing began at seven km/h (six km/h for women), and increased by 0.1 km/h every six seconds. The slope remained constant at 1%. The test ended when the subject could no longer run, and made a gesture with his/her hand; then, the recovery phase began at 4 km/h for three minutes, followed by rest for another two minutes. The tests were considered to be maximal and valid when they exceeded 85% of the theoretical maximum heart rate (220-age), and the respiratory quotient (RER) was greater than 1.15.

Throughout the stress test, the subjects breathed through a mask that was connected to a gas analyzer (Metalyzer 3b^®^, Cortex). Heart rate and electrocardiographic recordings were obtained with the Cardioline Cube^®^ device. All tests were carried out under similar environmental conditions (temperature 20–22 °C). The method used to determine the maximum oxygen uptake (VO_2_max) involved reaching the oxygen consumption plateau [27].

Anthropometric measurements were made by a sufficiently trained researcher, after calibrating all instruments. The height was determined for each subject (SECA^®^ tallimeter), and the total body mass, fat mass and musculoskeletal mass were measured with a bioimpedance scale (InBody 120^®^). The body mass index (BMI) was calculated from total mass/height^2^. Figure 2 shows the flowchart of the actions.

### 2.4. Statistical Analysis

After ruling out the presence of errors, the data were exported to the Statistical Package for Social Science (SPSSv.28^®^) to be analyzed. Prior to data analysis, the Shapiro–Wilk test and the Levene test were performed to determine the normal distribution of the variables, and the homogeneity of the variance. The quantitative variables were described in terms of the mean and standard deviation (SD). The qualitative variables were described using absolute frequency and percentage (%). The comparison of means between independent intergroup variables (males and females) was performed using Student’s *t*-test, and the comparison of the means of related variables was made with a paired *t*-test. The relationship between variables was studied using Pearson’s test, and effect size was analyzed with Cohen’s d (<0.2 small; 0.2–0.8 medium; >0.8 large).

Height, total body mass, percentage of fat mass, percentage of musculoskeletal mass and BMI were used as anthropometric independent variables. The independent ergospirometric variables were the resting heart rate, maximum HR, VO_2_max, maximum speed and maximum ventilation. As dependent variables, we used those provided by Max Pulse^®^ (Table 1) as well as its provided conversion variables (value before ST–value after ST). A minimum level of significance of *p* < 0.05 was established.

## 3. Results

In Table 2 we show the basic, anthropometric and body composition characteristics of the population, in addition to the maximum ST values, comparing men and women.

Through Max Pulse^®^, the following scores were awarded in the pre-effort assessment. They were compared by gender, and no significant differences were found between men and women (Table 3).

By analyzing the differences between the scores obtained before and after the stress test, we obtained the results shown in Table 4.

Correlations between the initial stress variables and age, hours of weekly sports practice, anthropometric values and body composition are shown in Table 5. Table 6 shows correlations between the stress variables and the ST variables.

The correlations that were significant between the conversions (value ST before S—value ST after) and the independent variables are shown in Table 7.

## 4. Discussion

The device that we have used studies variability in the heart rate, which allows for an analysis of the autonomic nervous system, and for an evaluation of physical/mental stress, through its own score, that relates to fatigue and electrocardiac stability. The PPG signal is a valuable source of physiological information, since it is influenced by the cardiovascular system and by the ANS, which can modify its response depending on the degree of stress [23]. We obtained information related to PRV at two different times, before and after intense physical exercise, in order to observe the changes produced by its action.

The intensity of the exercise was assessed on the basis of maximum speed and physiological variables (HR, VO_2_max and ventilation).

The software most often used for the calculation of HRV parameters is that derived from Kubios HRV^®^, which can obtain more than 30 parameters [15]. This information may be adequate for the investigation and analysis of an individual, but the information is cumbersome for uninitiated people, and complex to understand for many athletes or patients. For this reason, programs and apps that analyze the basic information, and express the results in a simple and immediate way, can be useful to assess large populations quickly and easily [26]. The use of algorithms, artificial intelligence (AI), and frequency domain analysis of HRV embedded in apps and devices provides easy interpretations of these data that can be used to improve the quality of life and health of patients and users [28]. Max Pulse^®^, like other modern devices, is based on the use of these technological advances for the interpretation of results and the automatic use of data, facilitating the investigations of researchers in the assessment and quantification of stress [29,30,31,32]. The novelty of this device is that it facilitates the interpretation of the data associated with the influence of HRV on the ANS, both in absolute values and when comparing between two different situations, in our case before and after intense exercise.

Plethysmography-based devices are becoming increasingly widespread, especially applications that are incorporated into phones, obtaining good correlations when compared to electrocardiograms [15]. Max Pulse^®^, like other applications, assesses PRV from the index finger. We consider that this location is valid for determinations at rest, but it is not valid for obtaining data during exercise. Other devices capable of obtaining the same information, but from the wrist, have recently been introduced on the market, although there are authors who indicate that it is less precise [18].

In our work, the score of the different variables has been obtained from the two classic measurement methods of HRV [3]. The temporal dimension from the calculation of the standard deviation of the heart rate of the RR intervals and the frequency dimension, quantifying the levels of each heart rate frequency. The analysis of each particular frequency, and separately, provides information on the sympathetic and parasympathetic systems [33]. The frequencies used to obtain the scores have been the usual ones in HRV studies [1]: total power (TP), very low frequency (VLF), low frequency (LF) and high frequency (HF). Although the protocol used by the device does not respect the recommendations of minimum measurement times. Based on the frequency values, the Max Pulse^®^ device performs the interpretations. Reduced TP: decreased ANS function, decreased regulatory capacity of organs, and decreased ability to cope with environmental changes. VLF Reduction: Decreased ability to regulate body temperature and hormone levels. Reduced LF: Loss of energy, fatigue, too little sleep, tiredness, and lethargy. Reduced HF: Chronic stress, aging, reduced electrical stability of the heart. Max Pulse^®^ graphically displays this information (Figure 3).

The validity of the use of digital plethysmography to assess stress has been previously verified by other authors [34]; thus, we consider its use to be acceptable. As expected, after performing a ST, an increase in heart rate is evident in both sexes, since the time elapsed between the end of the test and the measurement of PRV was insufficient to make a full recovery. After physical exercise, and while not returning to the heart rate prior to it, the values obtained by plethysmography indicated a persistence of the stress situation. In addition, the variables that increased their values were “physical stress” and “score stress”, and those that decreased, “ANS” and “Stress resistance”, were the expected ones.

On the other hand, in women, the score of “Mental Stress” and “ANS Balance” also decreased. All of these changes were consistent, and showed an increase in SNS activity that was associated with physical exercise [7]. After intense exercise, the overall stress score rose, specifically “physical stress”. We observed with this system that a decrease in HRV was associated with maximum exercise, which coincided with the research by Poehling and Llewellyn [35], who compared HRV between rest and exercise states. It appeared that at rest, the influence of the resting PNS contributed to a higher HRV, while the action of the SNS during maximal exercise led to a reduced HRV. Furthermore, they indicated that these effects may be related to the enhanced automaticity effects of norepinephrine acting at its B1 receptor sites in the heart. The global score of ANS activity decreased; this was due to the decrease in HRV. The changes were less evident in “mental stress”, which only decreased in women.

According to the study by Soltani et al., the intensity of an exercise program produces more benefits to improving the ANS than the volume of exercise [36]. These changes refer to training, not acute exercise, so they cannot be applied directly to our research. Thus, there are studies that show how physical training can decrease sympathetic activity and increase parasympathetic activity, decreasing HR at rest [37]. On the other hand, it has been observed that when exercise is performed at higher intensities, there is more release of norepinephrine and, therefore, greater sympathetic activity and more physiological stress [36]. Lee et al. recently found that both aerobic exercise and resistance training were effective for sympathetic nerve activities in middle-aged women, and that the effects on sympathetic and parasympathetic activities were greater for resistance training [38]. Bonet et al. also observed differences in stress that is associated with physical activity between active and sedentary people, finding that the impact of exercise was less in active people; however, we did not find this correlation [7]. This may be due to the fact that in our study population, there were no strict sedentary individuals, but active people with different levels of dedication.

On the other hand, an association between adiposity and HRV in the results of physical exercise programs was sought [39]. We found that before the stress test, the balance of the ANS showed a positive correlation with fat mass and the percentage of fat mass. Previously, Sztazjel et al. found that a decrease in HRV was associated with an increase in fat mass [40]. Triggiani el al. associated a decrease in HRV in young women with an increase in visceral fat mass [41]. Phoemsapthawee et al. suggested that obese sedentary young men achieved significant improvements in vagal activity, adiposity indices and aerobic fitness after exercise training; in addition, ANS balance was also positively related to mental stress [39].

ANS balance is calculated using the LF/HF ratio; an increase in this ratio expresses more sympathetic activity, and a decrease indicates more parasympathetic activity [38]. At rest, healthy people maintain a balance between the SNS and PNS. This relationship is altered by subjecting the subjects to a stressful situation, such as physical exercise.

The measurement of HRV before and after intense exercise has been used to study the risk of pathology and cardiovascular death. The research of Dewey et al. [42] is classic, in which they suggested that greater variability after exercise is associated with increased cardiovascular risk. They used an electrocardiogram; currently, this can assessed in a simpler way, using PPG.

The data provided by Max Pulse^®^ inform about the balance of the ANS in each individual. This balance is influenced, in addition to the variables we measured, by other situations and characteristics. Therefore, when analyzing the entire group, we were not able to find relationships between them. We did not find a direct relationship between the variables of the ST and the initial values of the variables that quantify stress. The stress score maintained a negative correlation only with the percentage of skeletal muscle mass. This supports the idea that increasing muscle mass through physical exercise can lower stress scores.

The ST acts as an acute exercise; it managed to modify the scores of almost all the stress variables provided by our device; however, no relationship was observed between these changes and the intensity values reached in the test. This is probably due to the fact that the test was a maximum measurement in all of the subjects; if we had performed a submaximal test, possibly some of its parameters would either have not been modified, or would have maintained a specific relationship with other variables. However, there are authors who indicate that changes in PRV are due to the mere fact of performing exercise, regardless of its intensity [35].

The score of the variable “resistance to stress” decreased significantly as a consequence of the ST. This can be associated with exhaustion after the test, and interpreted as that; after physical effort, the subjects were at a disadvantage to face a new situation. This change occurred inversely in the stress score, which increased significantly after the effort. Both changes occurred in a similar way in men and women, and can be used in the planning of physical exercise and rehabilitation programs.

Numerous studies have been published about the factors that influence the different HRV parameters, such as age [43], the amount of physical activity performed [14], or the time of day in which the exercise is done [44]. However, there is little research about the factors that influence the interpretation of the software and applications of specific devices, as has been our case. Holmes et al. studied it with applications on smartphones, analyzing the changes associated with strength exercises in the RMSSD, and looked for the validity of the results before and after a training session [45], while Singstad et al. studied it during rest, exercise and mental stress [46].

The main limitation of our study when using this device was that it did not provide data of the main HRV parameters (SDNN, RMSSD and frequency spectrum); thus, its information could not be compared directly with other applications, or with other monitoring systems that track PRV and HRV. With our design, we could not provide information on the validity of this equipment, but we could assess its ability to analyze the influence of exercise on its variables, and we observed that the data it provided were coherent and consistent. This allows for its practical application in informing athletes of their baseline situation of ANS balance and stress level, and how exercise influences their particular situation. If the person is too stressed, the sympathetic nervous system is dominant, or if the person is fatigued, the parasympathetic system is dominant. The information provided by the device can be obtained before and after training and rehabilitation sessions, in order to objectify acute changes, and to monitor recovery between sessions to help planning.

## 5. Conclusions

We conclude that performing intense exercise causes changes in the scores of variables that assess the balance of the ANS and the stress, as measured by a PPG-based device equipped with its own software. These changes are consistent with those detected with other technologies, appear in both genders, and are not related to body composition.

## Figures and Tables

**Figure 1 ijerph-19-16066-f001:**
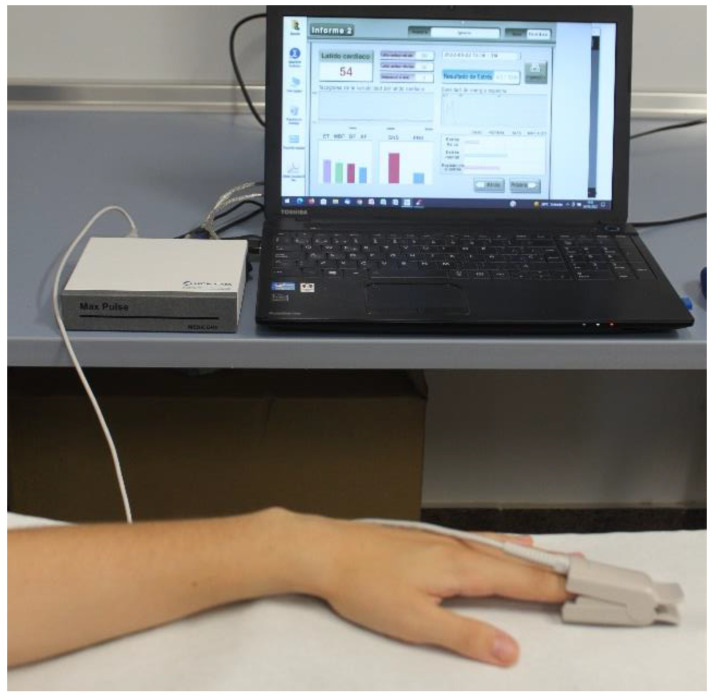
Max Pulse digital sensor and device.

**Figure 2 ijerph-19-16066-f002:**
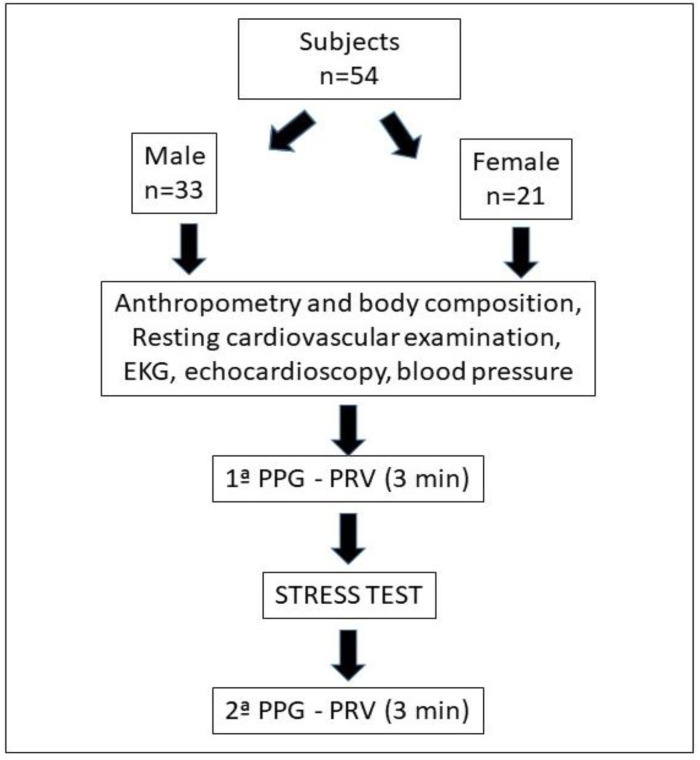
Flowchart of participants and procedures.

**Figure 3 ijerph-19-16066-f003:**
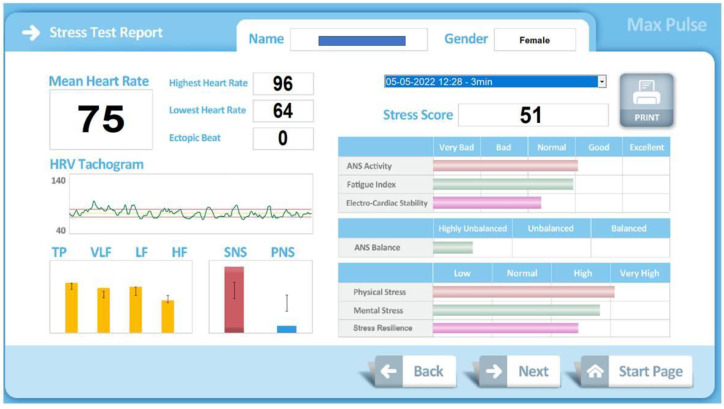
Graphic information provided by Max Pulse^®^.

**Table 1 ijerph-19-16066-t001:** Variables of autonomic nervous activity and stress.

Variable	Definition	Score Rating (Maximum 100 Points)
ANS Activity	It indicates the activity of the function of the autonomic nervous system and its regulatory competence.	Excellent (100–80), good (80–60), normal (60–40), bad (40–20), very bad (20–0)
Balance ANS ANS Balance	Indicates the balance between SNS and PNS. The normal ratio is 6:4.	Balanced, unbalanced, very unbalanced
Physical stress	It shows the level of stress in the physical state.	Very high (100–75), high (75–50), normal (50–25), low (25–0)
Mental stress	It shows the level of stress in the mental state.	Very high (100–75), high (75–50), normal (50–25), low (25–0)
Stress Resistance	It means the ability to sustain control stability in the face of stress.	Very high (100–75), high (75–50), normal (50–25), low (25–0)
Score Stress	Displays the stress score. The average score is 50, and a higher score means more stress. Less than 50 means better cardiovascular health.	Very high (100–75), high (75–50), normal (50–25), low (25–0)

**Table 2 ijerph-19-16066-t002:** Comparison between sexes of the basic characteristics of the population, and of the maximum values of the stress test (males = 33; females *n* = 21).

	Gender	Mean	SD	t Value	*p* Value	d Cohen’s
Age (Years)	Male	31.3	8.1	0.819	0.417	0.228
	Female	29.5	7.8			
Hours/Week	Male	5.1	3.2	2.814	0.007 *	0.768
	Female	2.9	2.1			
Height	Male	176.0	6.0	6.971	0.000 *	1.946
	Female	162.7	7.9			
Body mass (kg)	Male	74.3	10.6	4.646	0.000 *	1.297
	Female	60.0	11.7			
BMI (kg/m^2^)	Male	24.0	3.0	1.446	0.154	0.404
	Female	22.6	4.0			
Musculoskeletal mass (kg)	Male	34.9	4.1	10.029	0.000 *	2.800
	Female	23.4	4.1			
Fat mass (kg)	Male	12.9	5.2	−2.367	0.022 *	−0.661
	Female	17.2	8.3			
Fat mass (%)	Male	16.9	5.0	−7.052	0.000 *	−1.968
	Female	28.0	6.6			
Musculoskeletal mass (%)	Male	43.6	5.4	−0.853	0.398	−0.238
	Female	44.8	4.6			
Heart rate (bpm)	Male	186.1	10.8	0.103	0.919	0.029
	Female	185.8	8.0			
% Theorical heart rate	Male	97.8	3.7	−0.954	0.345	−0,266
	Female	98.9	4.8			
Velocity (km/h)	Male	14.2	2.7	−0.605	0.548	−0,169
	Female	14.7	3.4			
VO_2_max (mL/kg/min)	Male	43.9	8.4	−1.052	0.297	−0,294
	Female	46.5	9.3			
Ventilation (L/min)	Male	110.1	29.2	−0.028	0.977	−0.008
	Female	110.3	34.0			

VO_2_max (mL/kg/min): maximal oxygen consumption; BMI (kg/m^2^): body mass index; * *p* < 0.05: significant difference.

**Table 3 ijerph-19-16066-t003:** Max Pulse^®^ scores prior to the stress test.

*n* = Male/Female 33/21	Gender	Mean	SD	t Value	*p* Value
ANS	Male	7.8	1.1	−0.725	0.472
	Female	8.0	1.1		
ANS Balance	Male	58.9	20.0	−1.042	0.302
	Female	64.6	18.4		
Physical stress	Male	21.3	16.2	0.555	0.581
	Female	17.0	9.4		
Mental stress	Male	2.4	2.8	−0.541	0.591
	Female	2.9	3.2		
Stress Resistance	Male	64.3	17.7	−0.793	0.431
	Female	68.7	19.9		
Score Stress	Male	37.3	21,6	0.245	0.808
	Female	35.1	23.6		
HR (bpm)	Male	68.2	13.4	0.362	0.719
	Female	66.3	10.2		
Systolic blood pressure (mmHg)	Male	125.3	12.5	−0.322	0.749
	Female	126.4	12.0		
Diastolic blood pressure (mmHg)	Male	73.2	6.6	−1.244	0.219
	Female	77.0	15.1		

ANS: autonomic nervous system; ANS balance: autonomic nervous system balance; HR: heart rate.

**Table 4 ijerph-19-16066-t004:** Comparison, separated by gender, of differences in scores between after and before the ST.

Gender		POST ST		PRE ST		Mean Differences		t Paired		d
		Mean	SD	Mean	SD	Mean	SD	t Value	*p* Value	Cohen’s
Male (*n* = 33)	ANS	4.9	1.0	7.8	1.1	−2.9	1.7	−9.621	0.000 *	−1.556
	ANS Balance	53.4	20.4	58.9	20.0	−5.5	27.5	−1.122	0.271	−0.556
	Physical Stress	206.3	127.6	21.3	16.2	−85	132.4	7.124	0.000 *	0.841
	Mental Stress	1.8	1.8	2.4	2.8	−0.6	2.9	−1.177	0.249	−0.575
	Stress resistance	19.1	6.7	64.3	17.7	−45.3	18.4	−13.48	0.000 *	−3.180
	Score Stress	80.7	15.7	37.3	21.6	43.4	25.4	9.507	0.000 *	1.146
	Heart rate	101.1	15.3	68.2	13.4	32.9	15.3	12.01	0.000 *	1.504
Female (*n* = 21)	ANS	4.9	1.0	8.0	1.1	−3.1	1.4	−10.29	0.000 *	−3.049
	ANS Balance	49.4	20.5	64.6	18.4	−15.2	27.1	−2.566	0.018 *	−1.015
	Physical Stress	180.6	105.3	17.0	9.4	163	97.5	6.281	0.000 *	0.840
	Mental Stress	1.1	0.7	2.9	3.2	−1.8	3.4	−2.259	0.037 *	−0.992
	Stress resistance	18.4	5.0	68.7	19.9	−50.3	19.4	−11.61	0.000 *	−3.517
	Score Stress	80.8	7.5	35.1	23.6	45.7	26.3	7.758	0.000 *	1.024
	Heart rate	99.4	11.1	66.3	10.2	33.1	8.4	18.11	0.000 *	2.659

ANS: autonomic nervous system; ANS balance: autonomic nervous system balance; * *p* < 0.05: significant difference.

**Table 5 ijerph-19-16066-t005:** Correlations between initial stress variables and general variables.

Correlations		ANS	ANS Balance	Physical Stress	Mental Stress	Stress Resistance	Score Stress
Age (years)	r Pearson	−0.193	−0.192	0.111	−0.209	−0.345	0.262
	*p* Value	0.162	0.165	0.450	0.138	0.012 *	0.058
Hours/week	r Pearson	0.073	−0.088	−0.180	−0.065	0.062	−0.126
	*p* Value	0.600	0.526	0.215	0.647	0.662	0.369
Height (cm)	r Pearson	−0.010	−0.053	−0.005	−0.043	−0.166	0.037
	*p* Value	0.941	0.706	0.970	0.762	0.239	0.792
Body mass (kg)	r Pearson	−0.004	0.265	0.060	0.334	−0.059	0.030
	*p* Value	0.976	0.052	0.680	0.015 *	0.679	0.833
BMI (kg/m^2^)	r Pearson	−0.010	0.382	0.097	0.501	0.042	0.001
	*p* Value	0.942	0.004 *	0.508	0.000 *	0.770	0.992
Musculoskeletal mass (kg)	r Pearson	0.007	0.046	0.009	0.036	−0.081	0.001
	*p* Value	0.962	0.741	0.949	0.800	0.566	0.992
Fat mass (kg)	r Pearson	−0.016	0.416	0.098	0.527	0.032	0.045
	*p* Value	0.910	0.002 *	0.501	0.000 *	0.819	0.747
Fat mass (%)	r Pearson	−0.025	0.328	0.104	0.401	0.047	0.068
	*p* Value	0.860	0.015 *	0.478	0.003*	0.740	0.630
Musculoskeletal mass (%)	r Pearson	0.248	0.340	−0.106	0.338	0.211	−0.307
	*p* Value	0.071	0.012 *	0.470	0.014 *	0.132	0.025 *

BMI (kg/m^2^): body mass index; * *p* < 0.05: significant correlation.

**Table 6 ijerph-19-16066-t006:** Correlations between initial stress variables and differences after the ST.

Correlations		ANS	Balance ANS	Physical Stress	Mental Stress	Stress Resistance	Score Stress
Heart rate (bpm)	r Pearson	−0.055	−0.099	−0.005	−0.143	0.017	0.131
	*p* Value	0.695	0.479	0.970	0.310	0.907	0.351
% Theoretical heart rate	r Pearson	0.146	0.038	−0.148	0.089	0.221	−0.254
	*p* Value	0.291	0.783	0.312	0.532	0.115	0.066
Velocity (km/h)	r Pearson	0.072	0.243	0.048	0.343	−0.095	−0.126
	*p* Value	0.606	0.076	0.744	0.013 *	0.503	0.369
VO_2_max (mL/kg/min)	r Pearson	0.130	0.291	0.048	0.313	−0.024	−0.198
	*p* Value	0.348	0.033 *	0.746	0.024 *	0.867	0.156
Ventilation (L/min)	r Pearson	0.124	0.293	−0.050	0.343	−0.025	−0.154
	*p* Value	0.372	0.032 *	0.730	0.013 *	0.859	0.271

VO_2_max (mL/kg/min): maximal oxygen consumption; * *p* < 0.05: significant correlation.

**Table 7 ijerph-19-16066-t007:** Correlations between ST before—ST after differences and independent variables.

Correlations		Dif. ANS	Dif. Balance ANS	Dif. Physical Stress	Dif. Mental Stress	Dif. Stress Resistance	Dif. Score Stress
Age (years)	r Pearson	0.107	0.287	0.202	0.269	0.349	−0.267
	*p* Value	0.451	0.039 *	0.212	0.061	0.013 *	0.059
BMI (kg/m^2^)	r Pearson	0.005	−0.210	−0.220	−0.389	0.025	−0.040
	*p* Value	0.972	0.136	0.172	0.006 *	0.865	0.783
Fat mass (kg)	r Pearson	−0.093	−0.360	−0.177	−0.513	−0.013	−0.057
	*p* Value	0.510	0.009 *	0.275	0.000 *	0.929	0.689
Fat mass (%)	r Pearson	−0.085	−0.373	−0.162	−0.459	−0.060	−0.061
	*p* Value	0.550	0.007 *	0.317	0.001 *	0.678	0.673
Musculoskeletal mass (%)	r Pearson	−0.333	−0.376	0.090	−0.334	−0.301	0.330
	*p* Value	0.016 *	0.006 *	0.581	0.019 *	0.033 *	0.018 *
Heart rate (bpm)	r Pearson	−0.072	0.609	0.121	−0.096	0.032	0.343
	*p* Value	0.611	0.000 *	0.406	0.507	0.824	0.013 *
Velocity (km/h)	r Pearson	−0.238	0.003	−0.342	−0.021	0.196	0.133
	*p* Value	0.090	0.984	0.016 *	0.885	0.168	0.347
VO_2_max (mL/kg/min)	r Pearson	−0.323	0.060	−0.351	−0.082	0.230	0.129
	*p* Value	0.020 *	0.712	0.013 *	0.572	0.105	0.361
Ventilation (L/min)	r Pearson	−0.286	0.088	−0.347	−0.085	0.207	0.206
	*p* Value	0.040 *	0.588	0.014 *	0.558	0.145	0.143

* *p* < 0.05: significant correlation.

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
