# Peer review of "Influences of Intense Physical Effort on the Activity of the Autonomous Nervous System and Stress, as Measured with Photoplethysmography"

_ijerph, 2022, doi:10.3390/ijerph192316066_

Round 1

Reviewer 1 Report

The need to measure VO2max in relation to the objectives set in the study is not understood. There is no discussion on this matter either. For greater clarity in the results, you should consider removing them.

Author Response

The need to measure VO2max in relation to the objectives set in the study is not understood. There is no discussion on this matter either. For greater clarity in the results, you should consider removing them.

ANSWER: We welcome the reviewer's comments.

The measurement of VO2 max has been carried out to obtain parameters related to the intensity of the exercise performed and to relate them to the stress variables. We have seen (Table 6) that there is a significant (although weak) relationship with ANS Balance and mental stress scores. We have written a new paragraph in the discussion to influence this issue.

"The intensity of the exercise has been assessed by maximum speed and physiological variables (HR, VO2max and ventilation)".

Reviewer 2 Report

Minor observations

·        Although the article is clear, checking the (English) writing and grammar is advisable. In all sections, some writing mistakes are perceivable by a non-native English speaker (like me).

·        Method writing is a bit messy. It is suggested to organize this section as follows: Study design, study sample, measurement, experimental protocol, and statistical analysis.

Introduction

·         The focus of the study is not clear enough. Was the main subject of the study exploring the exercise-induced changes of stress, or was it to evaluate if the device is effective in identifying (by the stress score)  the exercise induces changes of stress? In my opinion, both approaches present substantial differences in study design, analysis, and conclusions. It suggests adding more antecedents to define the horizon of the study.

·         The punctuation provided by Max Pulse was succinctly mentioned in the introduction. It should add some empirical information about the utility, responsiveness, and validity of these punctuations against suitable reference standards.

·         A broad body of evidence shows the changes in HRV induced by exercise. What is new with studying the same phenomenon by Max Pulse? It should add the importance of studying the exercise-induced changes in autonomic balance and stress by Max Pulse.

·         Why did the authors decide to analyze differences between sexes and their association with body composition? It should be added this justification in the introduction. (As commentary: I understood this point after reading the whole text. I suggest improving the explanation that the study focuses on verifying if the results provided by Max Puls are consistent with the theory frame.

Methods

·         It is unclear how Max Pulse estimates balance ANS, physical stress, mental stress, etc. . It should be detailed how those indexes are derivated from the HRV. In addition, which HRV indexes were used for the stress calculations? Considering that a short pulse recording was analyzed (5 min), it is necessary to clarify which HRV variables were used. This is because HRV indexes' precision is tightly dependent on time recording. Eg., SDNN, LF, and VLF, among others, are not recommended in short-term HRV analysis.

·         In the method was written: "In the same way, the HRV was performed after the stress test." Whether HRV was performed, dependent or independent, it is necessary to provide how RR was recorded, the method of analysis, the interpolation methods, breathing control (yes or no), and the estimated HRV indexes.

·         How was the sample size calculated? This should be informed in statistical analysis.

·         It is not clear why the post-test measurement was 5-minutes after. Why not during the exercise test? Or immediately at the end of it?

RESULTS

·         Table 4 is a bit confusing. Why the mean differences were calculated as [pre-values] – [post-values]? Could it be post–pre values? This doubt flourishes, since ANS, in post-values, decreases with a positive mean difference. Contrary to physical stress, the post-test increased sustancially, but their difference is negative. The same occurs with other indexes. I suggest considering presenting the results' values and tendencies more intuitively.

·         The results shown in table 5 are related to objectives other than those presented in the introduction. How does the correlation analysis between baseline variables contribute to the study objectives? I suggest mentioning some of this background in the introduction.

·         In table 5, the correlation between ANS balance with Fat mass and musculoskeletal mass has the same tendency (positive correlation). What is that mean? May it be a contradiction?

·         Table 5 is too difficult to interpret due to the sense of correlations. Eg, Velocity should have a positive correlation with Dif. Physical stress (the more Velocity, the more physical stress, doesn't it?). Again, I suggest checking the sense of tendencies (Focus on table 4).

DISCUSSION

·         The following sentence was presented in the discussion: "the frequencies used to obtain the scores have been the usual ones in HRV studies, among them: VLF and LF". How was that possible with short-term HRV analysis? As mentioned above, some time domain HRV indexes, LF and VLF are not recommended to use with recordings shorter than 5 minutes. In addition, this information should be detailed in the methodology.

·         Despite that being an issue of the post-revision stage,  the quality of figure two is too low. I suggest adding another with a better resolution in the method section. Besides, the values exhibited may not be representative enough (See Welch's periodogram). I suggest checking this point.  

·         In my opinion, stress is not the target measurement of digital plethysmography. Instead, it is the pulse. In this context, HRV would be a variable derivated from the pulse. I suggest checking this argument.

·         To be precise, all SNA and HRV changes should be interpreted and analyzed as a post-exercise response. That is important because, according to how it was discussed, it looks like the HRV changes occurred during the exercise test. I suggest reviewing this point in the discussion.   

To contribute to this discussion, I believe the changes in HRV observed in the study represent an late stage of HR and HRV recovery, which has been demonstrated to be dependent on exercise intensity and duration. For that reason would be essential to know the values of HRV variables in addition to the stress values.

·         In the first paragraph of page 10, the authors mention several studies with contradictory results but without any explanation of this phenomenon. It should be added.

·         In the same page, there is a possible mistake. It is mentioned that the balance of the ANS has a negative correlation with fat mass and a positive with the percentage of muscle-skeletal mass. However, in table 5 the r of Fat mass is 0,328 and the r of Musculoesqueletal mass is 0,340. Please check it.

·         I suggest reviewing the second paragraph of page 10. It begins to analyze the association between HRV and physical exercise, the relationship between stress and ANS balance, and mental stress. The ideas seem to be unconnected.

·         The Study from Williamson-Reisdorf et al is not related to this study. I suggest reviewing that paragraph.

·          I suggest reviewing this paragraph (attached at the end of this point). First, It is focused on the relationships of baseline variables. However, this was not included in the objectives and hypotheses. Second: What did the authors mean by the highlighted sentence?: "The physical stress and mental stress scores help to determine the origin of the situation of excessive stress in each subject individually, but when analyzed as a group, it cannot be appreciated. We have not found a direct relationship with the variables of the ST and the initial values of the variables that quantify stress. The stress score maintains a negative correlation only with the skeletal muscle mass percentage. This supports the idea that increasing muscle mass through physical exercise can lower stress scores". Please check it.

·         In my opinion, the limitation is a considerable limitation that should also be added to the methodology. On the other hand, the authors mentioned that HR was measured before and after exercise. Is it possible to estimate HRV from these data? If it is possible, I recommend doing it because to support the conclusion; more empirical findings are needed.

·          

Author Response

Although the article is clear, checking the (English) writing and grammar is advisable. In all sections, some writing mistakes are perceivable by a non-native English speaker (like me).

ANSWER. We appreciate the comment. We have reviewed the manuscript and corrected the errors found.

  • Method writing is a bit messy. It is suggested to organize this section as follows: Study design, study sample, measurement, experimental protocol, and statistical analysis.

ANSWER. We appreciate the suggestion. We have reordered the subsections 2.1 Study design 2.2 Participants; 2.3. Measurements 2.4 Statistical analysis

Introduction

1.- The focus of the study is not clear enough. Was the main subject of the study exploring the exercise-induced changes of stress, or was it to evaluate if the device is effective in identifying (by the stress score) the exercise induces changes of stress? In my opinion, both approaches present substantial differences in study design, analysis, and conclusions. It suggests adding more antecedents to define the horizon of the study.

ANSWER: The study has been carried out to determine the changes produced by a stress test on the information provided by the device. It is not intended to evaluate the effectiveness of the device itself or to compare it with other means. To clarify this, a new phrase has been introduced in the design section of the study.

“With this we will obtain information about the scores prior to the exercise and after it”

2.- The punctuation provided by Max Pulse was succinctly mentioned in the introduction. It should add some empirical information about the utility, responsiveness, and validity of these punctuations against suitable reference standards.

ANSWER. The information we provide is that provided by the device manufacturer. It is a novel system, there are no references in terms of comparison with other devices.

3.-  A broad body of evidence shows the changes in HRV induced by exercise. What is new with studying the same phenomenon by Max Pulse? It should add the importance of studying the exercise-induced changes in autonomic balance and stress by Max Pulse.

ANSWER. We appreciate the comment. Indeed, there are numerous studies carried out to determine the changes induced by exercise in HRV. Our contribution is to carry out this assessment with a device that, instead of providing the data of the parameters of the variability, does so with a qualitative score of them.

We have included in the discussion a phrase that supports this idea.

“The novelty of this device is that it facilitates the interpretation of the data associated with the influence of HRV on the ANS, both in absolute values and when comparing between two different situations, in our case before and after intense exercise.”

4.- Why did the authors decide to analyze differences between sexes and their association with body composition? It should be added this justification in the introduction. (As commentary: I understood this point after reading the whole text. I suggest improving the explanation that the study focuses on verifying if the results provided by Max Pulse are consistent with the theory frame.

ANSWER. We appreciate the question. We decided to analyze the differences between sexes and body composition because there are previous studies that show that both situations influence HRV. We include in the introduction a sentence to clarify this topic with recent bibliographical references.

“In addition, other authors have observed that HRV is related to gender and body composition”.

DeWayne PW, Joseph N, Gerardo NM, Hill LK, Koenig J, Thayer JF. Gender Differences in Cardiac Chronotropic Control: Implications for Heart Rate Variability Research. Appl Psychophysiol Biofeedback. 2022; 47:65–75. doi.org/10.1007/s10484-021-09528-w

Chintala KK, Krishna BH, Reddy MN. Heart Rate Variability in Overweight Health Care Students: Correlation with Visceral Fat. JCDR. 2015. 9(1): CC06-CC08.doi: 10.7860/JCDR/2015/12145.5434

Methods

5.-  It is unclear how Max Pulse estimates balance ANS, physical stress, mental stress, etc. . It should be detailed how those indexes are derivated from the HRV. In addition, which HRV indexes were used for the stress calculations? Considering that a short pulse recording was analyzed (5 min), it is necessary to clarify which HRV variables were used. This is because HRV indexes' precision is tightly dependent on time recording. Eg., SDNN, LF, and VLF, among others, are not recommended in short-term HRV analysis.

ANSWER: We completely agree with the reviewer's commentary. As mentioned above and in the text of the work, the equations and variables that Max Pulse to determine their parameters are not published and are one of the aspects that we have criticized in the discussion. We have written that it constitutes the main limitation of our work.

6.-  In the method was written: "In the same way, the HRV was performed after the stress test." Whether HRV was performed, dependent or independent, it is necessary to provide how RR was recorded, the method of analysis, the interpolation methods, breathing control (yes or no), and the estimated HRV indexes.

ANSWER. Thank you for the observation. That phrase has been deleted. What has been valued has been the PRV through photoplestimography.

7.- How was the sample size calculated? This should be informed in statistical analysis.

ANSWER. A sample size has not been calculated as it is not intended to extrapolate the results to a population. It is a descriptive study of the subjects who came to our laboratory in a certain time (first semester of 2022).

“They came to our lab in the first half of 2022”.

8.- It is not clear why the post-test measurement was 5-minutes after. Why not during the exercise test? Or immediately at the end of it?

ANSWER.- The PRV is not determined during the effort because the heart rate changes due to the exercise and this variability is due to the effect of the exercise performed. After the effort, five minutes are left to stabilize the heart rate for recovery. The first three minutes, the athlete continues to monitor and perform an active recovery on the tapestry (stress test protocol). Subsequently, the mask is removed, transferred to the table and the device is placed. The time needed to make the second measurement is standardized to five minutes. With the same time for all participants.

RESULTS

9.- Table 4 is a bit confusing. Why the mean differences were calculated as [pre-values] – [post-values]? Could it be post–pre values? This doubt flourishes, since ANS, in post-values, decreases with a positive mean difference. Contrary to physical stress, the post-test increased sustancially, but their difference is negative. The same occurs with other indexes. I suggest considering presenting the results' values and tendencies more intuitively.

ANSWER. We agree with the reviewer that the differences can be raised in both directions: PRE-POST and POST-PRE. In any of them we can obtain negative and / or positive values in the variables, depending on whether it is a tendency to raise or lower the final values with respect to the initial ones. Absolute values do not change. We appreciate the comment that the POST-PRE form may be more intuitive for the reader and proceed to change the signs and order of the columns.

10.-  The results shown in table 5 are related to objectives other than those presented in the introduction. How does the correlation analysis between baseline variables contribute to the study objectives? I suggest mentioning some of this background in the introduction.

ANSWER. Our aim is to determine if the information provided by Max Pulse® on autonomic balance and stress is modified after intense physical exercise, determining if there is a relationship with body composition and also if there are differences with respect to gender. 

We consider that, before knowing what is the influence of body composition and other variables on changes, it is necessary to establish what are the values in the PRE situation and its relationships. That is, it is a previous step, first we know how the relationship is and then how it changes.

11.-  In table 5, the correlation between ANS balance with Fat mass and musculoskeletal mass has the same tendency (positive correlation). What is that mean? May it be a contradiction? Table 5 is too difficult to interpret due to the sense of correlations. Eg, Velocity should have a positive correlation with Dif. Physical stress (the more Velocity, the more physical stress, doesn't it?). Again, I suggest checking the sense of tendencies (Focus on table 4).

ANSWER. Indeed, it seems that these are contradictory data. Although significant, the value of the ratio (r) is low, so the importance is small. It leads us to think that there are other factors that can influence. For example, gender, or the number and characteristics of subjects (there is little dispersion of values in most variables).·         

DISCUSSION

12.- The following sentence was presented in the discussion: "the frequencies used to obtain the scores have been the usual ones in HRV studies, among them: VLF and LF". How was that possible with short-term HRV analysis? As mentioned above, some time domain HRV indexes, LF and VLF are not recommended to use with recordings shorter than 5 minutes. In addition, this information should be detailed in the methodology.

ANSWER: We welcome the feedback. These are the data provided by the manufacturer of the device and is the protocol of use. We add the observation in the discussion.

“Although the protocol used by the device does not respect the recommendations of minimum measurement times.”

13.- Despite that being an issue of the post-revision stage,  the quality of figure two is too low. I suggest adding another with a better resolution in the method section. Besides, the values exhibited may not be representative enough (See Welch's periodogram). I suggest checking this point.  

ANSWER. Thanks for the suggestion. A new image will be sent.

14.- In my opinion, stress is not the target measurement of digital plethysmography. Instead, it is the pulse. In this context, HRV would be a variable derivated from the pulse. I suggest checking this argument.

ANSWER. Effectively. We agree with the reviewer that plethysmography serves to assess the characteristics of the pulse. One of the characteristics is the variability of the pulse, equivalent to the variability of the heart rate and one of its applications is the quantification of stress.

15.- To be precise, all SNA and HRV changes should be interpreted and analyzed as a post-exercise response. That is important because, according to how it was discussed, it looks like the HRV changes occurred during the exercise test. I suggest reviewing this point in the discussion.

ANSWER. We agree with the reviewer that the variables have been measured before and after the exercise. Therefore, the cause of the changes has been exercise. The wording of the first paragraph of the introduction is modified to prevent it from being misinterpreted.

“We have obtained the information related to the PRV at two different times: before and after intense physical exercise, in order to observe the changes produced by its action”.

16.- To contribute to this discussion, I believe the changes in HRV observed in the study represent an late stage of HR and HRV recovery, which has been demonstrated to be dependent on exercise intensity and duration. For that reason would be essential to know the values of HRV variables in addition to the stress values.

ANSWER. Table 4 shows, in addition to the differences by the action of physical exercise, the initial and final values with the variables provided by the device. We cannot provide HRV's own variables because, as expressed in several workplaces, they are not available.

17.- In the first paragraph of page 10, the authors mention several studies with contradictory results but without any explanation of this phenomenon. It should be added.

ANSWER: Very grateful. We have added the explanation:  “This may be due to the fact that in our population there are no strict sedentary, but people with different dedication”.

18.- In the same page, there is a possible mistake. It is mentioned that the balance of the ANS has a negative correlation with fat mass and a positive with the percentage of muscle-skeletal mass. However, in table 5 the r of Fat mass is 0,328 and the r of Musculoesqueletal mass is 0,340. Please check it.

ANSWER. We welcome the observation. We corrected the error and improved the wording.

19.- I suggest reviewing the second paragraph of page 10. It begins to analyze the association between HRV and physical exercise, the relationship between stress and ANS balance, and mental stress. The ideas seem to be unconnected.

ANSWER. We have improved the wording. We have separated the paragraph in two. The phrase has been included.  “After physical exercise, and while not returning to the heart rate prior to it, the values obtained by plethysmography indicate the persistence of the stress situation”.

20.- The Study from Williamson-Reisdorf et al is not related to this study. I suggest reviewing that paragraph.

ANSWER. We appreciate the comment. We have changed the text and reference to better align it with the purpose of our study.  “The measurement of HRV before and after intense exercise has been used to study the risk of pathology and cardiovascular death. The work of Dewey et al. is classic, in which they suggested that greater variability after exercise is associated with increased cardiovascular risk. They used an electrocardiogram, currently it could be assessed, in a simpler way, by plethysmography”.

Dewey FE, Freeman JV, Engel G, Oviedo R, Abrol N, Ahmed N, Myers J,  Froelicher VF. Novel predictor of prognosis from exercise stress testing: Heart rate variability response to the exercise treadmill test. AHJ. 2007; 153(2):281-8. doi:10.1016/j.ahj.2006.11.001

21.-  I suggest reviewing this paragraph (attached at the end of this point). First, It is focused on the relationships of baseline variables. However, this was not included in the objectives and hypotheses. Second: What did the authors mean by the highlighted sentence?: "The physical stress and mental stress scores help to determine the origin of the situation of excessive stress in each subject individually, but when analyzed as a group, it cannot be appreciated. We have not found a direct relationship with the variables of the ST and the initial values of the variables that quantify stress. The stress score maintains a negative correlation only with the skeletal muscle mass percentage. This supports the idea that increasing muscle mass through physical exercise can lower stress scores". Please check it.

ANSWER: We appreciate the comment. The data provided by Maxpulse informs about the balance of the ANS of each individual. This balance is influenced, in addition to the variables we have measured, by other situations and characteristics. Therefore, when analyzing the entire group, we have not been able to find relationships between them. We improved the wording of the paragraph by including another sentence.

            “The data provided by Maxpulse informs about the balance of the ANS of each individual. This balance is influenced, in addition to the variables we have measured, by other situations and characteristics. Therefore, when analyzing the entire group, we have not been able to find relationships between them”.

  • In my opinion, the limitation is a considerable limitation that should also be added to the methodology. On the other hand, the authors mentioned that HR was measured before and after exercise. Is it possible to estimate HRV from these data? If it is possible, I recommend doing it because to support the conclusion; more empirical findings are needed.

ANSWER. We welcome the reviewer's opinions and comments. We believe that we have answered all of them and clarified the doubts, contributing to improve the manuscript. Thank you very much.

Reviewer 3 Report

In the abstract, it is suggested not to use acronyms or initials, but to develop the full expression of the meaning of the name to facilitate clarity of reading, and not to use initials in isolation such as "RR Intervals".  Be careful with spaces between words, just one. 

In 'Max Pulse' also 'Maxpulse' has been written without spaces.

 It is suggested to remove all brackets from the abstract, except those pertaining to statistical significance.

Author Response

We thank the reviewer for the time spent in valuing our work and for their comments and suggestions.

1.- In the abstract, it is suggested not to use acronyms or initials, but to develop the full expression of the meaning of the name to facilitate clarity of reading, and not to use initials in isolation such as "RR Intervals".  Be careful with spaces between words, just one. 

ANSWER: We appreciate the comment. We have removed unnecessary acronyms and initials. The wording of "RR Intervals" has been corrected.

2.- In 'Max Pulse' also 'Maxpulse' has been written without spaces.

ANSWER. We appreciate the observation and regret the error. They have been corrected, it is always written Max Pulse.

3.-  It is suggested to remove all brackets from the abstract, except those pertaining to statistical significance.

ANSWER. We appreciate the comment. Parentheses have been removed.

Round 2

Reviewer 2 Report

The article was substantially improved. Almost all observations were effectively corrected.

There are some minor issues that I suggest rechecking before the acceptance:

·         The treadmill model is not commonly presented in the abstract. All other devices should also be mentioned if the treadmill model is present. I suggest reviewing it.

·         In result section, the statement "autonomic nervous system decreased"… should be corrected. (ANS does not decrease… It is sympathetic or parasympathetic nerve activity that increases or decreases). Please check it.

·         (Regarding answer no 10) If the authors consider that the objective was determining if the information provided by Max Pulse is modified after intense physical exercise. Therefore, the study object was the device's responsiveness (a measurement property). For methodological rigor, this point should be mentioned either in the introduction or in the method. I suggest reviewing this recent paper by Mokkink et al. ( https://doi.org/10.1016/j.jclinepi.2021.06.002).

Author Response

We would like to thank the reviewer for his comments and contributions.

1.- The treadmill model is not commonly presented in the abstract. All other devices should also be mentioned if the treadmill model is present. I suggest reviewing it.

ANSWER: We have removed the model name.

2.- In result section, the statement "autonomic nervous system decreased"… should be corrected. (ANS does not decrease… It is sympathetic or parasympathetic nerve activity that increases or decreases). Please check it.

ANSWER. Thanks for the heads up. We complete the sentence: Autonomic nervous system activity and mental stress decrease significantly.

3.-   (Regarding answer no 10) If the authors consider that the objective was determining if the information provided by Max Pulse is modified after intense physical exercise. Therefore, the study object was the device's responsiveness (a measurement property). For methodological rigor, this point should be mentioned either in the introduction or in the method. I suggest reviewing this recent paper by Mokkink et al. ( https://doi.org/10.1016/j.jclinepi.2021.06.002).

ANSWER: We have added: All this is related to the responsiveness of the device.